# Geometrically Constrained Gaussian Splatting SLAM

## Abstract

3D Gaussian Splatting (3DGS) has emerged as a promising technique in SLAM due to its rapid and high-quality rendering capabilities. However, its reliance on discrete Gaussian ellipsoid primitives limits its effectiveness in capturing essential geometric features crucial for accurate pose estimation. To overcome this limitation, we propose a novel dense RGB-D SLAM system that integrates an implicit Truncated Signed Distance Function (TSDF) hash grid to constrain the distribution of Gaussian ellipsoids. This innovative approach enables precise estimation of the scene's geometric structure by smoothing the discrete Gaussian ellipsoids and anchoring them to the scene's surface. Acting as a low-pass filter, the implicit TSDF hash grid mitigates the inductive biases inherent in traditional 3DGS methods while preserving rendering quality. Our geometrically constrained map also significantly enhances generalization capabilities for depth estimation in novel views. Extensive experiments on the Replica, ScanNet, and TUM datasets demonstrate that our system achieves state-of-the-art tracking and mapping accuracy at speeds up to 30 times faster than existing 3DGS-based systems.

## 1 Introduction

Visual Simultaneous Localization and Mapping SLAM (VSLAM), which parallels human visual perception, has garnered significant attention within the research community. Although traditional VSLAM systems (Mur-Artal et al., 2015; Mur-Artal & Tardós, 2017; Qin et al., 2018; Campos et al., 2021) achieve high tracking accuracy, their map representations often fall short for downstream tasks. Recently, various VSLAM systems have adopted Neural Radiance Fields (NeRF) by (Mildenhall et al., 2020) or 3DGS by (Kerbl et al., 2023), both based on differentiable rendering, as mapping solutions due to their high-quality rendering capabilities. Compared to NeRF, 3DGS offers faster rendering speeds and higher-quality rendering results, making it more suitable for real-time applications. Recent studies have paved the way for integrating 3DGS into VSLAM.

A key challenge in previous works (Matsuki et al., 2024; Keetha et al., 2024) is utilizing Gaussian ellipsoids to accurately represent the geometric structure of the scene, which significantly influences the accuracy of pose estimation. Although discrete Gaussian ellipsoids provide high rendering quality, their discrete nature leads to poor representation of scene geometric structures, as 3D reconstruction from multi-views is an underconstrained problem (Barron et al., 2022; Yu et al., 2024). Therefore, the continuous, implicit representation of NeRF provides a potential solution to solve the inaccurate geometric representation of 3DGS.

Another challenge stems from the increasing number of Gaussian ellipsoids required as the scene expands, complicating their management and optimization(Deng et al., 2024). Since rendering an image requires only a subset of the total Gaussian ellipsoids, this highlights the importance of utilizing a submap of ellipsoids. Moreover, the strategy for densifying Gaussian ellipsoids is critical to system performance (Chen & Wang, 2024). Adding too many ellipsoids increases the computational burden while adding too few results in an inadequate scene representation.

To address these challenges, we propose a novel system that combines the strengths of implicit and explicit representations. Our approach optimizes an implicit multi-resolution hash encoding (Müller et al., 2022) to predict TSDF (Azinović et al., 2022) values, which are then converted into opacities for each Gaussian ellipsoid. This hybrid mapping imposes geometric constraints on the unstructured Gaussian ellipsoids, enhancing the learning and generalization of scene geometric structures while

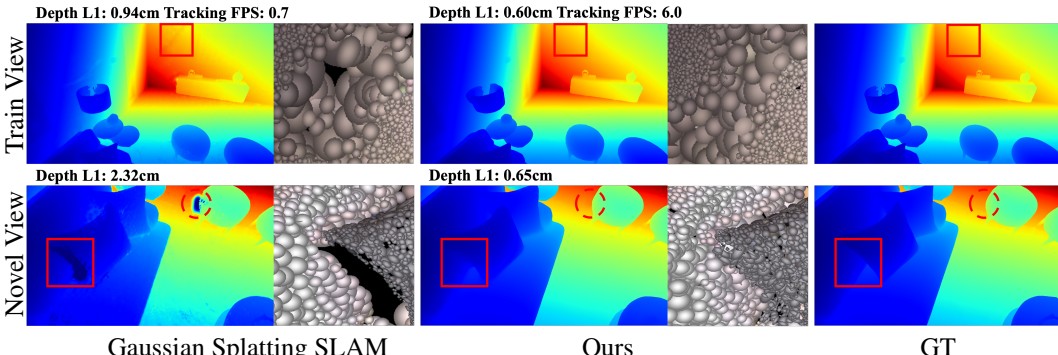

Figure 1: Comparison of geometric information rendering between our method and another method in train and novel views. In the depth maps estimated by each method, the Gaussian ellipsoid image on the right represents the area outlined by the red square in the depth map. The remaining red dashed lines in the depth map highlight the emphasized sections for comparison. It shows that our method provides a superior representation of scene geometry.

preserving rendering quality. As illustrated in Fig. 1, our system produces superior depth maps and local Gaussian ellipsoid results in both training and novel views compared to existing state-of-the-art 3DGS-based systems (Matsuki et al., 2024). Additionally, we mitigate the management and optimization issues of Gaussian ellipsoids by creating submaps for local tracking and gradually fixing certain ellipsoids. We dynamically add and remove Gaussian ellipsoids based on the cumulative opacity of each pixel and implicit TSDF predictions, reducing redundant ellipsoids and improving efficiency. Our system incorporates strategies related to keyframe selection, joint bundle adjustment, and a frontend-backend architecture to enhance robustness across various datasets.

Our main contributions can be summarized as follows:

- An enhanced and generalizable geometric TSDF hash grid constraint for the gaussian ellipsoids, which mitigating the inductive bias inherent in 3DGS and improves the precision of pose estimation and the operational speed of the system.
- A novel approach for dynamic management and optimization of Gaussian ellipsoids, seamlessly integrated into the SLAM system workflow. This method reduces the computational burden of the optimization process while preserving the accuracy and efficiency of mapping and tracking.
- We conducted extensive experiments across multiple datasets, demonstrating both the effectiveness and robustness of our method. Achieving state-of-the-art tracking and mapping accuracy, our system operates up to 30 times faster than existing 3DGS-based systems, setting a new benchmark for the community.

## 2 RELATED WORK

Here, we briefly introduce representative VSLAM systems. For a more detailed review, please refer to traditional SLAM surveys (Cadena et al., 2016; Macario Barros et al., 2022) and differentiable rendering-based SLAM survey Tosi et al. (2024).

**Traditional VSLAM.** Visual SLAM (VSLAM) systems can be categorized based on the sparsity or density of their map reconstructions. Sparse reconstruction systems (Davison et al., 2007; Mur-Artal et al., 2015; Engel et al., 2017; Mur-Artal & Tardós, 2017; Campos et al., 2021) functioned at higher speeds and primarily focused on camera tracking. However, the maps these systems produced often needed more detail for recognition tasks or other downstream applications due to their sparse nature. On the other hand, dense mapping VSLAM systems, while incurring higher computational costs due to dense reconstructions, have gained popularity in recent years for applications in Augmented Reality (AR) and robotics, where detailed environmental representation is essential. KinectFusion (Newcombe et al., 2011a), a real-time RGB-D SLAM algorithm for 3D reconstruction

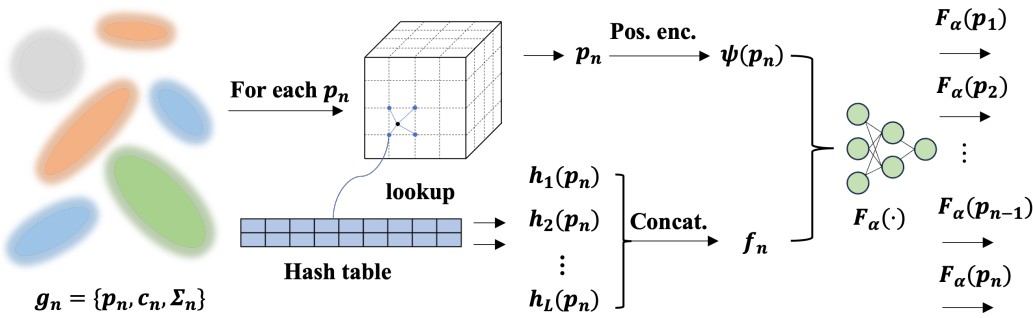

Figure 2: The map structure of our system. Each gaussian ellipsoid is characterized by its position, color, variance, and opacity, with opacity predicted based on the TSDF value. The implicit TSDF hash grid predicts a TSDF value for each gaussian ellipsoid at its respective location. Here $f_n$, $\psi(p_n)$ and $F_\alpha(\cdot)$ denotes the multi-resolution feature (Müller et al., 2022), position encoding (Müller et al., 2019; Wang et al., 2023) and decoder function for a certain point, respectively.

and surface mapping, encountered challenges such as accumulated drift over time. Pioneering direct methods like DTAM (Newcombe et al., 2011b) and MobileFusion (Ondrúška et al., 2015) utilized photometric error minimization to achieve dense reconstructions without relying on feature extraction. Semi-dense reconstruction approaches introduced by (Engel et al., 2014; Boikos & Bouganis, 2016; 2017) combined the advantages of both sparse and dense methods by reconstructing regions with high-information content. To enhance tracking and mapping accuracy, loop closure detection techniques were implemented by (Salas-Moreno et al., 2013; Kerl et al., 2013; Endres et al., 2013), mitigating drift by recognizing previously visited locations.

Moreover, with the recent advancements in deep learning, learning-based dense VSLAM methods (Ummenhofer et al., 2017; Tateno et al., 2017; Li et al., 2018; Kang et al., 2019; Yang et al., 2020; Li et al., 2020; Teed & Deng, 2021) emerged. We categorize these learning-based methods under the umbrella of traditional SLAM since their map representation and optimization strategies still follow previous SLAM systems.

**NeRF-based SLAM.** NeRF-based SLAM systems (Zhu et al., 2023a; Deng et al., 2023) represent a class of learning-based SLAM methods that utilize implicit map representations through volumetric rendering. One way to classify NeRF-based systems is by the method for pose estimation. Besides systems (Sucar et al., 2021; Yang et al., 2022; Li et al., 2023; 2024a) leveraging volumetric rendering to directly optimize camera poses, others (Kong et al., 2023; Chung et al., 2023; Rosinol et al., 2023; Zhang et al., 2023) incorporated traditional SLAM tracking modules to enhance performance. Another way to classify NeRF-based SLAM systems is by their map representations. Beyond the original NeRF (Mildenhall et al., 2020; Sucar et al., 2021), various structures have been explored, including Multi-MLP (Kong et al., 2023), Voxel Grid (Zhu et al., 2022; 2023b), Octree (Yang et al., 2022), Triplane (Chan et al., 2022; Johari et al., 2023), Hash Grid (Müller et al., 2022; Wang et al., 2023; Li et al., 2024a), and Neural Point Cloud (Sandström et al., 2023; Liso et al., 2024). Despite the rapid advancements in NeRF-based methods, their reliance on implicit map representations imposes limitations on rendering and training speeds.

Among NeRF-based SLAM systems, the early pioneers, iMAP (Sucar et al., 2021) and NICE-SLAM (Zhu et al., 2022) introduced NeRF into SLAM using MLP and Voxel Grid, respectively. Building on this, Co-SLAM (Wang et al., 2023) and ESLAM (Johari et al., 2023) adopted the InstantNGP (Müller et al., 2022) and Tri-plane (Chan et al., 2022), significantly improving both mapping and tracking accuracy, as well as speed. Recent systems like Go-SLAM (Zhang et al., 2023) and Loopy-SLAM (Liso et al., 2024) incorporated loop closure techniques, demonstrating superior performance over extended image sequences. However, with the emergence of 3DGS based on explicit Gaussian ellipsoids, the research focus of differentiable rendering-based SLAM has gradually shifted from NeRF to 3DGS, which enables fast and high-quality rendering.

**3DGS-based SLAM.** As an explicit map representation based on volumetric rendering, 3D Gaussian Splatting (3DGS) (Kerbl et al., 2023) has gained widespread use across various visual domains

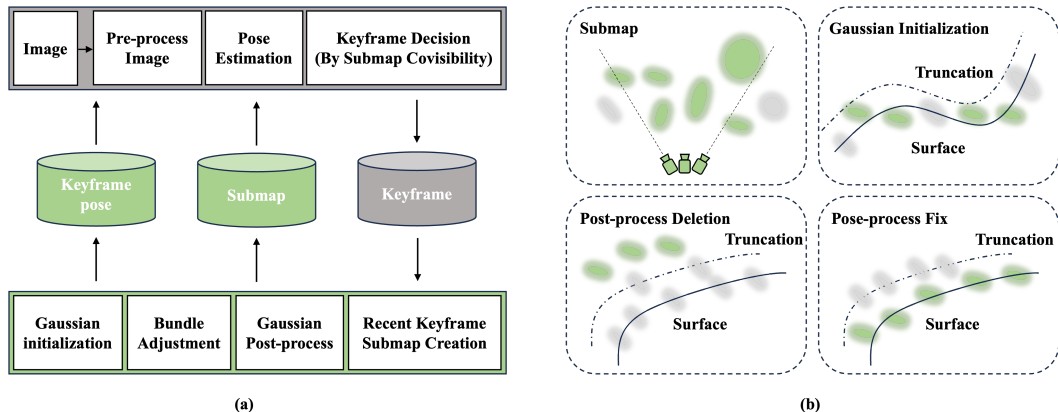

Figure 3: Overview of our system. (a) Structure of our system. It consists of two threads: mapping and tracking. These threads communicate by passing keyframes and gaussian submaps between them. (b) Management diagram for the gaussian ellipsoids. The green ellipsoids represent the ellipsoids that have undergone the corresponding operations.

since its introduction. Several studies have applied 3DGS in SLAM. Based on pose estimation methodologies, recent systems can be categorized into two types: systems using the traditional SLAM pose estimation method and systems leveraging 3DGS gradient backpropagation.

Systems (Ha et al., 2024; Sarikamis & Alatan, 2024; Hu et al., 2024b; Li et al., 2024b) that integrate precise pose estimates from traditional SLAM methods with carefully designed Gaussian ellipsoid processing strategies form one approach. For example, Gaussian-SLAM (Yugay et al., 2024) addresses the challenges of incorporating 3DGS into SLAM and utilizes DROID-SLAM (Teed & Deng, 2021) for pose estimation. Similarly, Photo-SLAM (Huang et al., 2024) employs ORB-SLAM3 (Campos et al., 2021) as the frontend tracking module and progressively refines the map in the backend. RTG-SLAM (Peng et al., 2024) combines frame-to-model ICP (Newcombe et al., 2011a) with ORB-SLAM2 (Mur-Artal & Tardós, 2017) backend optimization, simplifying the processing of Gaussian primitives and rendered depth. Alternatively, other systems (Hu et al., 2024a; Sun et al., 2024; Deng et al., 2024; Xu et al., 2024) focus on explicitly computing pose gradients by leveraging the fully differentiable nature of 3DGS. For instance, SplaTAM (Keetha et al., 2024) balances accuracy and speed by making specific assumptions about Gaussian primitives, while GS-SLAM (Yan et al., 2024) adopts a coarse-to-fine approach for pose optimization during tracking. GSS (Matsuki et al., 2024) introduces geometric verification and regularization techniques to resolve ambiguities in incremental 3D dense reconstruction.

Our work also leverages 3DGS gradient backpropagation. Different from previous pioneering systems, our system leverages available geometric information by utilizing an enhanced, generalizable geometric TSDF hash grid constraint for Gaussian ellipsoids. Furthermore, our system efficiently handles the adding, deleting, and optimizing Gaussian ellipsoids, minimizing redundant ellipsoids and improving overall optimization efficiency.

## 3 METHOD

Our task is to estimate camera poses $\{\mathbf{R}_i|\mathbf{t}_i\}_{i=1}^M$ from a set of sequential RGB-D frames $\{\mathbf{I}_i, \mathbf{D}_i\}_{i=1}^M$ with known camera intrinsics $\mathbf{K} \in \mathbf{R}^{3\times3}$, while simultaneously building a high quality dense map. We address the state estimation challenge by integrating a novel, generalizable geometric TSDF hash grid constraint and an advanced Gaussian ellipsoid processing algorithm. This hybrid approach mitigates the inductive biases present in previous 3DGS systems, improves the accuracy of pose estimation, and enhances the system's overall efficiency in mapping and tracking threads.

### 3.1 Geometrically Constrained Gaussian Ellipsoid

As shown in Fig. 2, our map is composed of explicit and implicit components. The implicit part is responsible for predicting the TSDF value of each explicit gaussian ellipsoid.

**Implicit TSDF hash grid representation.** We use a multi-resolution hash grid (Müller et al., 2022) to implicitly represent the TSDF value at each spatial point. For a point $\mathbf{x}_n$ in space, we have:

$$\mathbf{f}_n = \bigoplus_{l=1}^{L} h_l(\mathbf{x}_n), \tag{1}$$

where $L$ denotes the number of resolution levels in the hash grid. $h_l(\cdot)$ is the hash lookup and interpolation function that performs linear interpolation in the corresponding level for a certain point. $\mathbf{f}_n$ is the final feature obtained after concatenating each level's feature. Then, a two-layer MLP decodes $f_n$, resulting in the final TSDF value $s_n$.

TSDF has demonstrated powerful depth-constraining capabilities in NeRF-based SLAM. We incorporate TSDF into 3DGS-based SLAM to enforce depth constraints on gaussian ellipsoids and poses. We convert TSDF into opacity $\alpha_n$ using the following formula by (Or-El et al., 2022; Johari et al., 2023):

$$\alpha_n = 1 - e^{\frac{-\beta}{1+e^{\beta s_n}}}, \tag{2}$$

where $\beta$ is a parameter that controls the sharpness of the surface boundary.

**Explicit gaussian ellipsoids representation.** Thousands of gaussian ellipsoids render the final scene volumetrically. Each gaussian ellipsoid $\mathbf{g}_n$ consists of color $\mathbf{c}_n$, opacity $\alpha_n$, position $\mathbf{p}_n^w$, and variance (shape) $\mathbf{\Sigma}_n^w$. Since we aim not to produce high-quality images, we set the spherical harmonic order to zero, meaning each gaussian ellipsoid is solid-colored. We calculate the pixel location $\mathbf{p}_n^{pix}$ and the world and pixel variance $\mathbf{\Sigma}_n^w$, $\mathbf{\Sigma}_n^{pix}$ of the gaussian ellipsoid $\mathbf{g}_n$ on using the following formula:

$$\mathbf{p}_n^{pix} = M(\mathbf{T}_{cw}\mathbf{p}_n^w), \mathbf{\Sigma}_n^w = \mathbf{R}_n\mathbf{S}_n\mathbf{S}_n^T\mathbf{R}_n^T, \mathbf{\Sigma}_n^{pix} = \mathbf{J}\mathbf{R}_{cw}\mathbf{\Sigma}_W\mathbf{R}_{cw}^T\mathbf{J}^T, \tag{3}$$

where $\mathbf{R}_n \in \mathbf{R}^{3\times3}$ and $\mathbf{S}_n \in \mathbf{R}^{3\times3}$ are the rotation matrix and scale matrix of the gaussian ellipsoid. $\mathbf{J}$ and $\mathbf{R}_{cw}$ are the Jacobian of the projection function $M$ and the rotation component of the camera pose $\mathbf{T}_{cw}$, respectively.

**Volumetric rendering.** In accordance with the standard volumetric rendering process, for each pixel $i$, assuming the corresponding gaussian ellipsoid is arranged in ascending order of depths in the list $\mathcal{G}_i$, we can calculate the color $c_i$, depth $d_i$, and cumulative opacity $o_i$ for that pixel:

$$\mathbf{c}_i = \sum_{n\in\mathcal{G}_i} \mathbf{c}_n\alpha_n \prod_{j=1}^{i-1}(1-\alpha_j), d_i = \sum_{n\in\mathcal{G}_i} p_{n,z}^c\alpha_n \prod_{j=1}^{i-1}(1-\alpha_j), o_i = \sum_{n\in\mathcal{G}_i} \alpha_n \prod_{j=1}^{i-1}(1-\alpha_j), \tag{4}$$

where $p_{n,z}^c$ is the depth of the gaussian ellipsoid on the camera coordinate. The aforementioned depth rendering formula is applicable to both gaussian ellipsoids and the implicit hash grid. For implicit rendering, simply replace $p_{n,z}^c$ and $\alpha$ with the depth and opacity of the spatial points sampled along the sampled ray (Mildenhall et al., 2020).

### 3.2 Hybrid objective functions

**Implicit loss function.** Unlike the unstructured gaussian ellipsoids training in explicit methods, training the implicit hash grid requires handling the depth surface and space to ensure proper convergence of the entire implicit space. We first randomly sample a set number of pixels. For each pixel, we uniformly sample $N_u$ points along the ray from the optical center to the object surface and $N_d$ around the surface $[d_r - d_t, d_r + d_t]$. where $d_r$ is the intersection point's depth between object surface depth along the ray from the optical center to the selected pixel and $d_t$ is a hyperparameter that defines the sampling density. Following the practice in (Azinović et al., 2022; Johari et al., 2023), We look up the hash grid for each spatial point to obtain its TSDF value and then apply a loss to it, which can be divided into two cases:

- For spatial points within the truncation region $T$, we have:

$$\mathcal{L}_{in} = \frac{1}{|R|} \sum_{r \in R} \frac{1}{|P_r^{in}|} \sum_{p \in P_r^{in}} (z_p + s_p \cdot T - d_r)^2 . \tag{5}$$

- For spatial points outside the truncation region T, we have:

$$\mathcal{L}_{out} = \frac{1}{|R|} \sum_{r \in R} \frac{1}{|P_r^{out}|} \sum_{p \in P_r^{out}} (s_p - 1)^2 , \tag{6}$$

where $s_p \in [-1, 1]$ is the TSDF value of the sampled point and $R$ is the set of sampled pixels. $P_r^{in}$ and $P_r^{out}$ denote the points along the ray in or beyond the truncation region $T$. For points within the truncation region $T$, the closer a point is to the surface, the closer its TSDF value is to zero. Points inside the surface have negative TSDF values.

**Explicit loss function.** The Gaussian Splatting code implemented in CUDA renders the depth map and color map for a specific camera pose. We calculate their L1 loss concerning ground truth values:

$$L_c = \|D_{render} - D_{gt}\|_1 , L_d = \|I_{render} - I_{gt}\|_1 . \tag{7}$$

Additionally, we calculate a regularization loss for each gaussian ellipsoid to limit the ellipsoid's size in the third dimension (Matsuki et al., 2024) and constrain them near the depth surface, which is not constrained by the two-dimensional image:

$$L_r = \sum_{n=1}^{|\mathcal{G}|} \|s_n\|_1 + \left\| \mathbf{S}_n - \tilde{\mathbf{S}}_n \cdot \mathbf{1} \right\|_1 , \tag{8}$$

where $\tilde{\mathbf{S}}_n$ is the mean of the ellipsoid's scale $\mathbf{S}_n$. In explicit training, a single pixel's computed loss corresponds to multiple gaussian ellipsoids, creating a one-to-many constraint. However, in implicit training, each point forms an individual constraint on the implicit hash grid after sampling spatial points along the pixel ray. This explains why implicit training converges quickly and enforces stronger constraints.

## 3.3 THE PROPOSED SLAM SYSTEM

As shown in Fig. 3 (a), we divide our SLAM system into mapping and tracking threads. The mapping thread is responsible for the joint optimization of keyframe poses and the hybrid map (Fig. 3 (b)), while the tracking thread handles tracking the current frame's pose using the submap and determining keyframes.

### 3.3.1 MAPPING

We iteratively optimize our hybrid map and the poses of selected keyframes. Our mapping loss is composed of the following components:

$$L_{map} = \lambda_c L_c + \lambda_d L_d + \lambda_{\bar{d}} L_{\bar{d}} + \lambda_r L_r + \lambda_{in} L_{in} + \lambda_{out} L_{out}, \tag{9}$$

where $\lambda_{(\cdot)}$ represents the weight for each loss and $L_{\bar{d}}$ denotes the L1 loss of depth for implicit rendering. In implicit rendering, we render the depths of selected pixel points by sampling pixels and spatial points as previously described and then calculate the implicit depth L1 loss.

**Bundle adjustment.** During mapping, we optimize the pose concurrently. We select the top $N_k$ most relevant frames from the previous keyframe pool based on the current keyframe's pose, optimizing both the map and the poses.

**Gaussian ellipsoid initialization.** We initiate gaussian ellipsoids according to two criteria. For areas where the cumulative opacity is less than $\tau_o$ or where the depth significantly exceeds the actual depth, we initialize the relevant gaussian ellipsoids at the depth location. Additionally, after a certain number of iterations, we sample spatial points within a small random area and use our implicit map to predict their TSDF. For every spatial point whose TSDF is less than $\tau_s$, we search its nearest gaussian ellipsoid and calculate the distance $d$. We initialize a new gaussian ellipsoid on the location of the spatial point with $d$ larger than $\tau_d$.

Figure 4: Qualitative comparison of rendering images from different methods.

**Gaussian submap.** As the number of keyframes increases and the map expands, the number of gaussian ellipsoids also increases. Passing all gaussian ellipsoids without restriction for the tracking thread would result in increasingly longer rendering times per frame. Therefore, the mapping thread continuously bundles the co-visible gaussian ellipsoids from the most recent $N$ keyframes into a gaussian submap, passed to the tracking thread for tracking purposes. This prevents excessive growth in the number of gaussian ellipsoids to some extent.

**Gaussian post-process.** After a specified number of optimizations, we conduct post-processing on the gaussian ellipsoids in the map.

- Fixing Gaussian Ellipsoids: We track the number of optimizations $t_n$ and the average gradient $Grad_n$ for each gaussian ellipsoid. After a certain number of iterations, we fix the gaussian ellipsoids based on the following criteria:

$$F(g_n) = (t_n > \tau_t)(Grad_n < \tau_g)(\|s_n\| < \tau_{s1})(M_n > 0.99), \tag{10}$$

  where $M_n$ is the average cumulative opacity of all pixels when rendering the gaussian ellipsoid within the current keyframe window. Once fixed, the state variables, except for the opacity of this ellipsoid, no longer participate in further optimization.
- Deleting Gaussian Ellipsoids: We remove ellipsoids whose $\|s_n\|$ is above a certain threshold, effectively deleting ellipsoids too far from the object's surface.

### 3.3.2 TRACKING

**Pose estimation.** The submap in the tracking thread consists of implicit and explicit components. However, for each gaussian ellipsoid, we do not need to repeatedly calculate its opacity since the map is not optimized in the tracking thread. During tracking, we use the losses from explicit and implicit maps to iteratively optimize the pose via backpropagation.

$$L_{track} = \lambda_c L_c + \lambda_d L_d + \lambda_{\bar{d}} L_{\bar{d}} + \lambda_{in} L_{in} + \lambda_{out} L_{out}. \tag{11}$$

**Keyframe decision.** We determine whether to create a new keyframe based on two metrics—the proportion of pixels with a cumulative opacity less than $\tau_m$ and the intersection-over-union (IoU) ratio of the number of gaussian ellipsoids used to render the current frame versus those used for the previous keyframe.

## 4 EXPERIMENTAL RESULTS

### 4.1 SETUP

**Baselines.** We benchmark our method against two open-sourced 3DGS-based methods, which leverage 3DGS gradient backpropagation: GSS (Matsuki et al., 2024),SplaTAM (Keetha et al., 2024); three NeRF-based VSLAM methods: NICE-SLAM (Zhu et al., 2022), ESLAM (Johari et al., 2023) and Co-SLAM (Wang et al., 2023) and three classic methods: Kintinuous (Newcombe et al., 2011a), BAD-SLAM (Schops et al., 2019), ORB-SLAM2 (Mur-Artal & Tardós, 2017). Comparisons with other methods based on 3DGS can be found in the appendix.

Table 1: Mapping and Tracking Results of Replica.

| Method | Metric | Room0 | Room1 | Room2 | office0 | office1 | office2 | office3 | office 4 | Avg. |
|---|---|---|---|---|---|---|---|---|---|---|
| NICE-SLAM | Depth L1 [cm] ↓ | 1.83 | 1.41 | 2.22 | 1.45 | 1.64 | 2.71 | 2.17 | 2.10 | 1.94 |
| | PSNR ↑ | 22.12 | 22.47 | 24.52 | 29.07 | 30.34 | 19.66 | 22.23 | 24.49 | 24.42 |
| | SSIM ↑ | 0.69 | 0.76 | 0.81 | 0.87 | 0.89 | 0.80 | 0.80 | 0.86 | 0.81 |
| | LPIPS ↓ | 0.33 | 0.27 | 0.21 | 0.23 | 0.18 | 0.24 | 0.21 | 0.20 | 0.23 |
| | ATE RMSE [cm] ↓ | 1.64 | 2.08 | 1.80 | 1.23 | 0.79 | 1.69 | 3.90 | 2.77 | 1.98 |
| Co-SLAM | Depth L1 [cm] ↓ | 1.05 | 0.85 | 2.37 | 1.24 | 1.48 | 1.86 | 1.66 | 1.54 | 1.50 |
| | PSNR ↑ | 27.27 | 28.45 | 29.06 | 34.14 | 34.87 | 28.43 | 28.76 | 30.91 | 30.24 |
| | SSIM ↑ | 0.91 | 0.91 | 0.93 | 0.96 | 0.97 | 0.94 | 0.94 | 0.90 | 0.94 |
| | LPIPS ↓ | 0.32 | 0.29 | 0.27 | 0.21 | 0.20 | 0.26 | 0.23 | 0.24 | 0.25 |
| | ATE RMSE [cm] ↓ | 0.70 | 1.09 | 1.21 | 0.56 | 0.60 | 2.08 | 1.58 | 0.71 | 1.07 |
| ESLAM | Depth L1 [cm] ↓ | 0.86 | 0.88 | 1.18 | 0.77 | 1.22 | 1.06 | 1.02 | 1.10 | 1.01 |
| | PSNR ↑ | 25.32 | 27.77 | 29.08 | 33.71 | 30.20 | 28.09 | 28.77 | 29.71 | 29.08 |
| | SSIM ↑ | 0.86 | 0.90 | 0.93 | 0.96 | 0.92 | 0.94 | 0.95 | 0.95 | 0.93 |
| | LPIPS ↓ | 0.31 | 0.30 | 0.25 | 0.18 | 0.23 | 0.24 | 0.20 | 0.20 | 0.25 |
| | ATE RMSE [cm] ↓ | 0.70 | 0.70 | 0.51 | 0.56 | 0.54 | 0.57 | 0.71 | 0.62 | 0.62 |
| GSS | Depth L1 [cm] ↓ | 1.01 | 0.73 | 0.78 | 0.87 | 0.54 | 2.47 | 3.42 | 1.87 | 1.47 |
| | PSNR ↑ | 32.94 | 35.81 | 36.03 | 40.07 | 41.18 | 35.64 | 35.26 | 33.79 | 36.34 |
| | SSIM ↑ | 0.93 | 0.95 | 0.96 | 0.97 | 0.97 | 0.96 | 0.95 | 0.93 | 0.95 |
| | LPIPS ↓ | 0.07 | 0.07 | 0.07 | 0.06 | 0.05 | 0.06 | 0.06 | 0.10 | 0.07 |
| | ATE RMSE [cm] ↓ | 0.76 | 0.37 | 0.23 | 0.66 | 0.72 | 0.30 | 0.19 | 1.46 | 0.58 |
| SplaTAM | Depth L1 [cm] ↓ | 0.54 | 0.47 | 0.61 | 0.39 | 0.30 | 0.71 | 1.41 | 1.39 | 0.73 |
| | PSNR ↑ | 32.80 | 33.89 | 35.2 | 38.2 | 39.1 | 31.9 | 29.70 | 31.81 | 34.11 |
| | SSIM ↑ | 0.98 | 0.97 | 0.98 | 0.98 | 0.97 | 0.97 | 0.95 | 0.95 | 0.97 |
| | LPIPS ↓ | 0.07 | 0.10 | 0.08 | 0.09 | 0.09 | 0.10 | 0.12 | 0.15 | 0.10 |
| | ATE RMSE [cm] ↓ | 0.31 | 0.40 | 0.29 | 0.47 | 0.27 | 0.29 | 0.32 | 0.55 | 0.36 |
| Ours | Depth L1 [cm] ↓ | 0.65 | 0.39 | 0.68 | 0.37 | 0.41 | 0.75 | 1.00 | 0.95 | 0.65 |
| | PSNR ↑ | 32.17 | 34.62 | 35.46 | 40.91 | 39.71 | 34.40 | 33.09 | 34.05 | 35.55 |
| | SSIM ↑ | 0.98 | 0.98 | 0.98 | 0.99 | 0.98 | 0.98 | 0.96 | 0.97 | 0.98 |
| | LPIPS ↓ | 0.07 | 0.09 | 0.07 | 0.07 | 0.07 | 0.08 | 0.07 | 0.09 | 0.08 |
| | ATE RMSE [cm] ↓ | 0.27 | 0.30 | 0.34 | 0.22 | 0.27 | 0.28 | 0.37 | 0.29 | 0.29 |

Our method achieved SOTA tracking and mapping results at speeds **eight** times faster than GSS (Matsuki et al., 2024) and **30** times faster than SplaTAM (Keetha et al., 2024).

**Datasets.** Following previous literature (Zhu et al., 2022; Matsuki et al., 2024), we tested our method on three datasets. We quantitatively evaluate the reconstruction and tracking quality on eight synthetic scenes from Replica (Straub et al., 2019). We also evaluate the tracking results on six scenes from ScanNet (Dai et al., 2017) and three scenes from TUM RGB-D (Sturm et al., 2012) datasets.

**Metrics.** For reconstruction quality, we report the standard photometric rendering quality metrics, including PSNR, SSIM, Depth L1, and LPIPS, evaluated following the method used in SplaTAM (Keetha et al., 2024). For tracking quality, we utilize the ATE RMSE (cm) metric (Sturm et al., 2012) for camera tracking evaluation.

**Implementation details.** We run our system on a desktop PC with an Intel Core i7-12700 CPU and a NVIDIA RTX 3090 GPU. We modified the CUDA-based implementation of 3DGS to support pose gradient backpropagation and depth rendering while all other code is implemented in PyTorch. Further experimental details can be found in the appendix.

## 4.2 MAIN RESULTS

**Tracking results.** In Tab. 1, Tab. 2 and Tab. 3, we present the tracking results on three representative datasets, with each result obtained by averaging five random runs. We use three colors— first , second , and third —to rank the performance in each scene. On the synthetic Replica dataset

Table 2: Results of Tum Dataset.

| Methods | fr1/desk | fr2/xyz | fr3/office | Avg. |
|---------|----------|---------|------------|------|
| Kintinuous | 3.70 | 2.90 | 3.00 | 3.20 |
| BAD-SLAM | 1.70 | 1.10 | 1.70 | 1.50 |
| ORB-SLAM2 | 1.60 | 0.40 | 1.00 | 1.00 |
| NICE-SLAM | 2.85 | 1.84 | 2.95 | 2.55 |
| Co-SLAM | 2.44 | 1.71 | 2.46 | 2.20 |
| ESLAM | 2.54 | 1.09 | 2.47 | 2.03 |
| GSS | 1.52 | 1.58 | 1.65 | 1.58 |
| SplaTAM | 3.35 | 1.24 | 5.16 | 3.25 |
| Ours | 1.44 | 1.01 | 1.53 | 1.33 |

Table 3: Results of Scannet Dataset.

| Methods | 0000 | 0059 | 0106 | 0169 | 0181 | 0207 | Avg. |
|---------|------|------|------|------|------|------|------|
| NICE-SLAM | 12.0 | 14.0 | 7.9 | 10.9 | 13.4 | 6.2 | 10.7 |
| Co-SLAM | 7.1 | 11.2 | 9.3 | 5.8 | 11.6 | 7.1 | 8.7 |
| ESLAM | 7.3 | 8.5 | 7.5 | 6.5 | 9.0 | 5.7 | 7.4 |
| GSS | 9.6 | 6.2 | 7.1 | 10.7 | 18.2 | 7.5 | 9.8 |
| SplaTAM | 12.8 | 10.1 | 17.7 | 12.1 | 11.1 | 7.5 | 11.9 |
| Ours | 5.6 | 9.1 | 6.8 | 5.9 | 9.6 | 7.0 | 7.3 |

(Straub et al., 2019), our tracking results surpassed those of SOTA methods. Our method also demonstrated significantly faster speeds than 3DGS-based methods, as detailed in Tab. 4. On the more challenging real-world dataset, ScanNet (Dai et al., 2017), our method achieved the fastest speed (Tab. 4) and outperformed most of the other methods in tracking accuracy. We also tested tracking accuracy on the TUM dataset (Sturm et al., 2012), adding classical SLAM methods for comparison. The results show that our method ranks among the SOTA for learning-based methods, surpassing the classic method ORB-SLAM2 in one scene and performing comparably in the other two. This demonstrates the effectiveness and robustness of our approach.

**Reconstruction results.** Like other methods, we also evaluated the reconstruction performance of each method on the Replica dataset (Straub et al., 2019), detailed in Tab. 1. Our reconstruction metrics surpassed most methods. When narrowing the comparison to 3DGS-based methods, given that the reconstruction quality based on 3DGS is already high, all methods performed comparably (Fig. 4) except for the Depth L1 metric. Our method inherits the advantages of NeRF's continuous representation and the rapid rendering capabilities of 3DGS. Therefore, in terms of the Depth L1 metric, our method surpasses the results of GSS (Matsuki et al., 2024) in most scenes at speed eight times faster than GSS and achieves comparable results to SplaTAM (Keetha et al., 2024) at speed 30 times faster.

### 4.3 RUNTIME ANALYSIS

We compared the running FPS, model parameter, and rendering speeds of different methods on the Replica (Straub et al., 2019) and ScanNet (Dai et al., 2017) datasets, as detailed in Tab. 4. The model parameter is defined as the storage space required to represent the scene, and the values for Replica's room0 and ScanNet's scene0000 are calculated to be filled in the table. It was observed that previous 3DGS-based methods significantly lagged in terms of FPS, whereas our method substantially increased the running speed of 3DGS-based approaches. Regarding model parameters, our method is slightly higher than other GSS (Matsuki et al., 2024) due to its commendable performance on novel depth views.

Regarding rendering speed, once training is complete, our method can transform our map into an explicit 3DGS map through a one-time query of each gaussian ellipsoid's opacity. Therefore, our method's rendering speed is comparable to other methods. This balance highlights the efficiency and effectiveness of our approach in leveraging the strengths of NeRF (Mildenhall et al., 2020) and 3DGS (Kerbl et al., 2023).

### 4.4 ABLATIONS

**Hybrid map representation.** Fig. 1 demonstrates the contribution of our hybrid map representation to depth estimation. Thanks to the continuous depth fitting provided by the hash grid, the depth maps we rendered ultimately show slightly better performance than 3DGS-based map representation on train views and significantly outperform 3DGS-based map representation on novel views. This highlights the effectiveness of integrating continuous depth modeling into our system, enhancing accuracy and robustness across viewing angles.

Table 4: Time and Memory Analysis.

| Methods | FPS↑ | | Model Param.↓ | | Render FPS↑ | |
|---|---|---|---|---|---|---|
| | Replica | ScanNet | Replica | ScanNet | Replica | ScanNet |
| NICE-SLAM | 0.9 | 0.7 | 40.9M | 88.7MB | 0.2 | 0.2 |
| Co-SLAM | 17.1 | 6.4 | 24.1MB | 46.2MB | 2.7 | 2.8 |
| ESLAM | 12.1 | 4.1 | 27.9MB | 68.2MB | 2.2 | 2.2 |
| GSS | 0.7 | 2.3 | 24.2MB | 5.6MB | 558 | 641 |
| SplaTAM | 0.2 | 0.2 | 243.4MB | 156.4MB | 96 | 104 |
| Ours | 6.0 | 6.8 | 34.6MB | 7.5MB | 504 | 632 |

Table 5: Ablation on Depth L1 (cm).

| View Direction | w/o. HM | ours-full |
|---|---|---|
| Train View | 1.24 | 0.65 |
| Novel View | 3.96 | 1.07 |

Table 6: Ablation on RMSE (cm).

| Metric | w/o. HM | w/o. KS | w/o. BA | ours-full |
|---|---|---|---|---|
| RMSE | 39.1 | 6.7 | 9.5 | 5.6 |
| FPS | 6.3 | 6.6 | 6.5 | 6.8 |

We also conducted quantitative tests to assess the impact of using the hybrid map on the RMSE of tracking and the Depth L1 of rendered depth maps in both train and novel views. Tab. 5 tests Depth L1 of Replica (Straub et al., 2019) room0, while Tab. 6 tests RMSE of ScanNet (Dai et al., 2017) scene0000. It is evident that the Hybrid Map (HM) significantly improves pose estimation accuracy when operating at nearly the same speed. This analysis helps us understand the effectiveness of HM in enhancing the precision of tracking and depth estimation across different viewing scenarios.

**System ablation.** Tab. 6 also showcases our ablation studies on keyframe selection (KS) and BA. Except for the variables involved in the ablation, all other parameters were kept consistent to ensure a fair comparison of the effects of these different factors on system performance. It can be observed that all three main strategies effectively enhanced the tracking accuracy. For more ablation studies, please refer to the appendix.

## 5 CONCLUSION

We propose a 3DGS-based RGB-D SLAM system with a hybrid map representation. Our system combines the strengths of both implicit and explicit map types—leveraging the continuous geometric constraints from implicit maps alongside the rapid, high-quality rendering capabilities of explicit maps—this results in state-of-the-art reconstruction and tracking accuracy and enhanced generalization across novel views. Our refined strategies for processing gaussian ellipsoids, selecting keyframes, and BA significantly boost our SLAM system's speed and accuracy. Extensive and meticulous experiments corroborate the efficacy of our approach.

Our work utilizes complementary fusion methods between explicit discrete 3DGS and implicit continuous NeRF, exploring a new way for accurately representing scene geometric information. Looking ahead, we aim to further enhance the scalability of our system for larger and more complex environments by exploring advanced geometric constraints in expansive scenes. Additionally, we plan to investigate the integration of efficient loop closure detection with 3DGS to improve the robustness and accuracy of SLAM systems, particularly in large-scale and dynamic environments. These future directions will ensure our approach remains at the SLAM research and applications forefront.

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

## A    IMPLEMENTATION DETAILS

The experiments were conducted on a desktop platform with an Intel Core i7-12700 CPU and an NVIDIA RTX 3090 GPU. For the explicit map optimization, we computed the Jacobian of each Gaussian ellipsoid concerning the pose being optimized, leveraging the CUDA implementation from Gaussian Splatting SLAM (Matsuki et al., 2024). To enable joint optimization with the implicit map, we manually implemented the Jacobian for the pose perturbations within the implicit map. The learning rate for the Gaussian ellipsoid features was set to 0.0025, while the learning rates for the scale and quaternions were set to 0.001. The learning rate for the position was determined as 0.000016 multiplied by the scene radius. For the implicit optimization, we used two-layer fully connected networks with a hidden feature size of 32 as decoders for predicting the TSDF. The Hash Grid level was set to 16 for the Replica dataset (Straub et al., 2019), with learning rates for the decoder and grid set to 0.01. The optimization learning rates for rotational and translational poses were set to 0.001. For the Replica dataset (Straub et al., 2019), we set $N_u$ to 32 and $N_d$ to 11. For the ScanNet dataset (Dai et al., 2017), $N_u$ was set to 96 and $N_d$ to 21. On the TUM dataset (Sturm et al., 2012), $N_u$ was set to 64 and $N_d$ to 21.

In the tracking thread for implicit maps, we use a constant velocity assumption to compute the initial pose of the tracking frame. We iterate 15 times, sampling 2000 pixels in each iteration. In the mapping thread, the number of points sampled on the rays corresponding to pixels varies across different datasets. For the Replica (Straub et al., 2019) dataset and ScanNet(Dai et al., 2017) dataset, we sample 4000 pixels each time. For the TUM dataset, we sample 2000 pixels each time.

When calculating reconstruction metrics, we followed the methodology of SplaTAM (Keetha et al., 2024), which involves computing relevant reconstruction metrics (PSNR, SSIM, LPIPS, Depth L1) every five frames. We refer to the metrics calculated in this manner as "Train View" metrics. Our ablation studies calculate reconstruction metrics under "Novel View" conditions. We efficiently generated depth maps from new poses using the Replica dataset's ground truth mesh files. We selected 100 images from these new poses to test the depth estimation in Novel View settings. This demonstrated the continuity and generalizability of the implicit TSDF Hash Grid regarding depth estimation. When computing poses, we adhere to the methodology used by Co-SLAM (Wang et al., 2023). For non-keyframes, we estimate and store their relative poses to adjacent keyframes. We store their estimated poses for keyframes and optimize them within the backend using joint Bundle Adjustment (BA). In the final calculation of the Root Mean Square Error (RMSE), we reconstruct the poses of all non-keyframes based on the poses of keyframes and their relative poses. We then calculate the final RMSE metric based on these reconstructed poses.

## B    JACOBIAN WITH RESPECT TO THE POSE MATRIX

Referring to (Matsuki et al., 2024), we provided the Jacobian for the pose during the rendering process. However, unlike (Matsuki et al., 2024), in addition to the explicit part (3DGS) contributing to the pose, we also provided the Jacobian for the implicit part (TSDF hash grid) concerning the pose to jointly optimize the pose using both explicit and implicit components. For the implicit part, while most previous works optimize the quaternion corresponding to the pose, we derived the Jacobian under left perturbation.

We define variables $\boldsymbol{\xi} = [\boldsymbol{\rho}^T, \boldsymbol{\phi}^T]^T \in \mathfrak{se}(3)(\boldsymbol{\phi} \in \mathfrak{so}(3), \boldsymbol{\rho} \in R^3)$ and $\boldsymbol{T}_{cw} \in SE(3)$. The gradient concerning $\boldsymbol{T}_{cw} \in SE(3)$ can be transformed into the gradient concerning the unconstrained variable $\boldsymbol{\xi}$. We define $\boldsymbol{a}^\wedge$ as the skew-symmetric transformation of $\boldsymbol{a} \in R^3$, and $\exp(\boldsymbol{a}^\wedge)$ as the corresponding exponential map. Without causing confusion, for $\boldsymbol{\xi} \in R^6$, we can also define $\boldsymbol{\xi}^\wedge$:

$$\boldsymbol{a}^\wedge = \begin{bmatrix} 0 & -a_z & a_y \\ a_z & 0 & -a_x \\ -a_y & a_x & 0 \end{bmatrix}, \quad \boldsymbol{\xi}^\wedge = \begin{bmatrix} \boldsymbol{\phi}^T & \boldsymbol{\rho} \\ \boldsymbol{0}^\top & 0 \end{bmatrix}, \tag{12}$$

$$\exp(\boldsymbol{a}^\wedge) = \sum_{n=0}^{\infty} \frac{1}{n!} (\boldsymbol{a}^\wedge)^n. \tag{13}$$

For each Gaussian ellipsoid involved in rendering, its contribution to the final pose $\boldsymbol{T}_{cw}$ gradient consists of two parts:

$$\frac{\partial \boldsymbol{p}^c}{\partial \boldsymbol{T}_{cw}}, \frac{\partial \boldsymbol{R}_{cw}}{\partial \boldsymbol{T}_{cw}}, \tag{14}$$

where $\boldsymbol{p}^c = \boldsymbol{T}_{cw}\boldsymbol{p}^w$ is the camera coordinate of the gaussian ellipsoid. We write it this way to emphasize that the 3DGS rendering process transforms Gaussian ellipsoids from the world coordinate system to the camera coordinate system.

For each spatial point $\boldsymbol{p}^c$ sampled along the ray from the optical center to the sampled pixel, its contribution to the final pose $\boldsymbol{T}_{cw}$ gradient is:

$$\frac{\partial \boldsymbol{p}^w}{\partial \boldsymbol{T}_{cw}}, \tag{15}$$

where $\boldsymbol{p}^w = \boldsymbol{T}_{wc}\boldsymbol{p}^c$ is the world coordinate of the sampled spatial point. We write it this way to emphasize that the implicit sampling process transforms points from the camera coordinate system to the world coordinate system, which is an inverse transformation compared with the explicit Gaussian ellipsoid transformation.

In order to account for the effects of these two inverse transformations when optimizing the pose, we need to analyze them under the same perturbation model. For simplicity, we adopt the left perturbation model 14 from (Matsuki et al., 2024), so we only need to compute the Jacobian in Equ. 15.

## B.1 Implicit Part's Jacobian

According to Equ.15, we need to compute the Jacobian of $\boldsymbol{p}^w$ with respect to $\boldsymbol{T}_{cw}$ (corresponding to $\boldsymbol{\xi}$) under left perturbation of $\delta\boldsymbol{\xi}$, which requires some transformations to achieve:

$$
\begin{aligned}
\frac{\partial \left(\boldsymbol{T}_{wc}\boldsymbol{p}^c\right)}{\partial \delta\boldsymbol{\xi}} &= \lim_{\delta\boldsymbol{\xi}\to 0} \frac{\left[\exp\left(\delta\boldsymbol{\xi}^\wedge\right)\exp\left(\boldsymbol{\xi}^\wedge\right)\right]^{-1}\boldsymbol{p}^c - \exp(\boldsymbol{\xi}^\wedge)^{-1}\boldsymbol{p}^c}{\delta\boldsymbol{\xi}} \\
&= \lim_{\delta\boldsymbol{\xi}\to 0} \frac{\exp\left(-\boldsymbol{\xi}^\wedge\right)\left(\boldsymbol{I}-\delta\boldsymbol{\xi}^\wedge\right)\boldsymbol{p}^c - \exp\left(-\boldsymbol{\xi}^\wedge\right)\boldsymbol{p}^c}{\delta\boldsymbol{\xi}} \\
&= \lim_{\delta\boldsymbol{\xi}\to 0} \frac{-\exp\left(-\boldsymbol{\xi}^\wedge\right)\delta\boldsymbol{\xi}^\wedge\boldsymbol{p}^c}{\delta\boldsymbol{\xi}} \\
&= \lim_{\delta\boldsymbol{\xi}\to 0} \frac{\begin{bmatrix} \boldsymbol{R}_{wc} & \boldsymbol{t}_{wc} \\ \boldsymbol{0}^\top & 1 \end{bmatrix}\begin{bmatrix} -\delta\boldsymbol{\phi} & -\delta\boldsymbol{\rho} \\ \boldsymbol{0}^\top & 0 \end{bmatrix}\boldsymbol{p}^c}{\delta\boldsymbol{\xi}} \\
&= \lim_{\delta\boldsymbol{\xi}\to 0} \frac{\begin{bmatrix} -\boldsymbol{R}_{wc}\delta\boldsymbol{\phi}^\wedge\boldsymbol{p}^c - \boldsymbol{R}_{wc}\delta\boldsymbol{\rho} \\ 1 \end{bmatrix}}{\delta\boldsymbol{\xi}} \\
&= \lim_{\delta\boldsymbol{\xi}\to 0} \frac{\begin{bmatrix} -\boldsymbol{R}_{wc}\delta\boldsymbol{\phi}^\wedge\boldsymbol{R}_{c\omega}\boldsymbol{R}_{\omega c}\boldsymbol{p}^c - \boldsymbol{R}_{wc}\delta\boldsymbol{\rho} \\ 1 \end{bmatrix}}{\delta\boldsymbol{\xi}} \\
&= \lim_{\delta\boldsymbol{\xi}\to 0} \frac{\begin{bmatrix} -\left(\boldsymbol{R}_{wc}\delta\boldsymbol{\phi}\right)^\wedge\boldsymbol{R}_{wc}\boldsymbol{p}^c - \boldsymbol{R}_{wc}\delta\boldsymbol{\rho} \\ 1 \end{bmatrix}}{\delta\boldsymbol{\xi}} \\
&= \lim_{\delta\boldsymbol{\xi}\to 0} \frac{\begin{bmatrix} \left(\boldsymbol{R}_{wc}\boldsymbol{p}^c\right)^\wedge\boldsymbol{R}_{wc}\delta\boldsymbol{\phi} - \boldsymbol{R}_{wc}\delta\boldsymbol{\rho} \\ 1 \end{bmatrix}}{[\delta\boldsymbol{\rho}^T, \delta\boldsymbol{\phi}^T]^T} \\
&= \begin{bmatrix} -\boldsymbol{R}_{wc} & \left(\boldsymbol{R}_{wc}\boldsymbol{p}^c\right)^\wedge\boldsymbol{R}_{\omega c} \\ \boldsymbol{0}^T & \boldsymbol{0}^T \end{bmatrix}.
\end{aligned}
\tag{16}
$$

The transition from the first to the second line uses Equ. 13 while neglecting higher-order terms. The transition from the sixth to the seventh line makes use of $\boldsymbol{R}\boldsymbol{t}^\wedge\boldsymbol{R}^T = (\boldsymbol{R}\boldsymbol{t})^\wedge$, where $R$ is assumed to be an orthogonal matrix. The transition from the seventh to the eighth line makes use of $\boldsymbol{a}^\wedge\boldsymbol{b} = -\boldsymbol{b}^\wedge\boldsymbol{a}$, where $\boldsymbol{a}, \boldsymbol{b} \in R^3$.

## B.2 EXPLICIT PART'S JACOBIAN

For completeness, we also present the explicit Jacobian introduced by (Matsuki et al., 2024), which applys a left perturbation $\delta\boldsymbol{\xi}$ to compute the Jacobian with respect to $\boldsymbol{\xi}$:

$$
\begin{aligned}
\frac{\partial(\boldsymbol{T}_{cw}\boldsymbol{p}^w)}{\partial\delta\boldsymbol{\xi}} &= \lim_{\delta\boldsymbol{\xi}\to 0} \frac{\exp\left(\delta\boldsymbol{\xi}^\wedge\right)\exp\left(\boldsymbol{\xi}^\wedge\right)\boldsymbol{p}^w - \exp\left(\boldsymbol{\xi}^\wedge\right)\boldsymbol{p}^w}{\delta\boldsymbol{\xi}} \\
&= \lim_{\delta\boldsymbol{\xi}\to 0} \frac{\left(\boldsymbol{I}+\delta\boldsymbol{\xi}^\wedge\right)\exp\left(\boldsymbol{\xi}^\wedge\right)\boldsymbol{p}^w - \exp\left(\boldsymbol{\xi}^\wedge\right)\boldsymbol{p}^w}{\delta\boldsymbol{\xi}} \\
&= \lim_{\delta\boldsymbol{\xi}\to 0} \frac{\delta\boldsymbol{\xi}^\wedge \exp\left(\boldsymbol{\xi}^\wedge\right)\boldsymbol{p}^w}{\delta\boldsymbol{\xi}} \\
&= \lim_{\delta\boldsymbol{\xi}\to 0} \frac{\begin{bmatrix} \delta\boldsymbol{\phi}^\wedge & \delta\boldsymbol{\rho} \\ \boldsymbol{0}^{\mathrm{T}} & 0 \end{bmatrix} \begin{bmatrix} \boldsymbol{R}_{cw}\boldsymbol{p}^w + \boldsymbol{t} \\ 1 \end{bmatrix}}{\delta\boldsymbol{\xi}} \\
&= \lim_{\delta\boldsymbol{\xi}\to 0} \frac{\begin{bmatrix} \delta\boldsymbol{\phi}^\wedge(\boldsymbol{R}_{cw}\boldsymbol{p}^w + \boldsymbol{t}) + \delta\boldsymbol{\rho} \\ \boldsymbol{0}^{\mathrm{T}} \end{bmatrix}}{[\delta\boldsymbol{\rho}, \delta\boldsymbol{\phi}]^{\mathrm{T}}} \\
&= \begin{bmatrix} \boldsymbol{I} & -(\boldsymbol{R}_{cw}\boldsymbol{p}^w + \boldsymbol{t})^\wedge \\ \boldsymbol{0}^{\mathrm{T}} & \boldsymbol{0}^{\mathrm{T}} \end{bmatrix}.
\end{aligned} \tag{17}
$$

The transition from the first line to the second line makes use of Equ. 13, while neglecting higher-order terms.

For the rotational component $\boldsymbol{R}_{cw}$, we only need to calculate its Jacobian with respect to the component $\boldsymbol{\phi}$ in $\boldsymbol{\xi} = [\boldsymbol{\rho}, \boldsymbol{\phi}]$.

$$
\begin{aligned}
\frac{\partial\boldsymbol{R}_{cw}}{\partial\delta\boldsymbol{\phi}} &= \lim_{\delta\boldsymbol{\phi}\to 0} \frac{\exp\left(\delta\boldsymbol{\phi}^\wedge\right)\exp\left(\boldsymbol{\phi}\right) - \exp\left(\boldsymbol{\phi}\right)}{\delta\boldsymbol{\phi}} \\
&= \lim_{\delta\boldsymbol{\phi}\to 0} \frac{\left(\boldsymbol{I}+\delta\boldsymbol{\phi}^\wedge\right)\exp\left(\boldsymbol{\phi}\right) - \exp\left(\boldsymbol{\phi}\right)}{\delta\boldsymbol{\phi}} \\
&= \lim_{\delta\boldsymbol{\phi}\to 0} \frac{\delta\boldsymbol{\phi}^\wedge}{\delta\boldsymbol{\phi}}\exp\left(\boldsymbol{\phi}\right) \\
&= \begin{bmatrix} -\boldsymbol{R}_{cw:,1}^\wedge \\ -\boldsymbol{R}_{cw:,2}^\wedge \\ -\boldsymbol{R}_{cw:,3}^\wedge \end{bmatrix},
\end{aligned} \tag{18}
$$

where $\boldsymbol{R}_{cw:,1}$ is the ith column of the matrix.

After calculating the Jacobian of each Gaussian ellipsoid and the spatial points with respect to the pose left perturbation $\delta\boldsymbol{\xi}$, summing these Jacobians yields the final Jacobian with respect to $\delta\boldsymbol{\xi}$.

## C  MORE ABLATIONS

### C.1  ABLATION ON POSE CONVERGENCE

In Fig. 5, we present line charts showing the changes in the difference between the estimated pose and the ground truth pose with increasing iterations for both Gaussian Splatting SLAM and our system. (a) illustrates the changes at the beginning of both systems. Since both systems have more than 1000 iterations during initialization, they learn accurate geometric representations, and the initial pose estimates are close to the true values. Both systems converge to a low pose error, with our system benefiting from the geometric constraints of TSDF, resulting in faster convergence. (b) shows the changes in pose error for both systems after running for some time. Our system's initial pose estimates are much lower than those of Gaussian Splatting SLAM after running for a while. This is due to the fast convergence brought by the geometric constraints of the TSDF grid for each frame, reducing cumulative error. Additionally, to control variables, we set the initial pose estimates of Gaussian Splatting SLAM to those of our system (yellow line). It can be observed that after 20

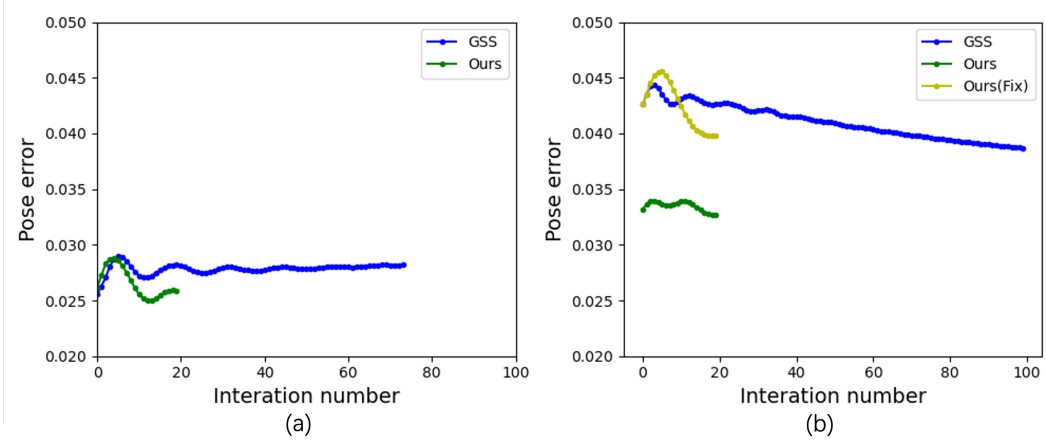

Figure 5: Comparison of pose convergence speed across different methods.



Table 7: Ablation study on Loss.

| Metric | w/o. $L_{in}$&$L_{out}$ | w/o. $L_r$ | ours-full |
|--------|------|------|------|
| RMSE | 27.1 | 5.8 | 5.6 |

Table 8: Ablation study on GPS.

| Metric | w/o. GF | w/o. GS | ours-full |
|--------|------|------|------|
| RMSE | 5.6 | 5.5 | 5.6 |



iterations, our system converges to the position equivalent to Gaussian Splatting SLAM after 60 iterations.

## C.2 ABLATIONS ON LOSS

We conducted ablation studies on the ScanNet scene0000 to evaluate the effects of different loss functions and Gaussian Processing Strategies (GPS), detailed in Tab. 7 and Tab. 8 "GF" represents fixed Gaussian ellipsoids. At the same time, "GS" denotes Gaussian submap. Although these strategies did not significantly improve the RMSE, they accelerated the optimization speed of Gaussian ellipsoids.

## D MORE RESULTS

### D.1 STATISTICAL SIGNIFICANCE

We conducted five runs to obtain the average values for our performance metrics. To assess the statistical significance of these results, we also calculated the standard deviation of the tracking and reconstruction metrics for each scene, as shown in Tab. 9, Tab. 10 and Tab. 11.

### D.2 MORE RESULTS ON RECENT OPEN-SOURCED 3DGS-BASED SYSTEMS

We compared several open-source 3DGS SLAM systems (Huang et al., 2024; Ha et al., 2024; Peng et al., 2024) based on traditional SLAM modules for tracking, which mainly focus on algorithms of optimizing the Gaussian ellipsoids. We note that since the pose provided by traditional SLAM modules is highly accurate and well-developed, our method falls short in pose tracking and speed compared to these methods. However, we believe that the pose gradient optimization approach based on 3DGS still has potential for development. Its advantage over traditional methods lies in providing continuous multi-view optimization capabilities, and it is more intuitive and aligned with human understanding than traditional feature-based methods. Tab. 12 presents the comparison of related metrics. Our method does not outperform SLAM systems based on traditional tracking modules on the Replica dataset, but it shows certain advantages on the TUM dataset. This is because the Replica dataset is synthetic, with highly accurate depth data, while the TUM dataset contains

Table 9: Statistical Significance Analysis on the Tum dataset

| Methods | fr1/desk | fr2/xyz | fr3/office |
|---------|----------|---------|------------|
| Ours | 0.08 | 0.04 | 0.11 |

Table 10: Statistical Significance Analysis on the Replica dataset

| Method | Metric | Room0 | Room1 | Room2 | office0 | office1 | office2 | office3 | office 4 |
|--------|--------|-------|-------|-------|---------|---------|---------|---------|----------|
| | Depth L1 [cm] ↓ | 0.06 | 0.04 | 0.04 | 0.03 | 0.06 | 0.04 | 0.02 | 0.03 |
| | PSNR ↑ | 0.41 | 0.56 | 0.14 | 0.48 | 0.08 | 0.31 | 0.77 | 0.71 |
| Ours | SSIM ↑ | 0.01 | 0.01 | 0.01 | 0.01 | 0.01 | 0.01 | 0.01 | 0.02 |
| | LPIPS ↓ | 0.01 | 0.01 | 0.01 | 0.01 | 0.01 | 0.02 | 0.01 | 0.02 |
| | ATE RMSE [cm] ↓ | 0.03 | 0.02 | 0.04 | 0.01 | 0.03 | 0.02 | 0.05 | 0.02 |

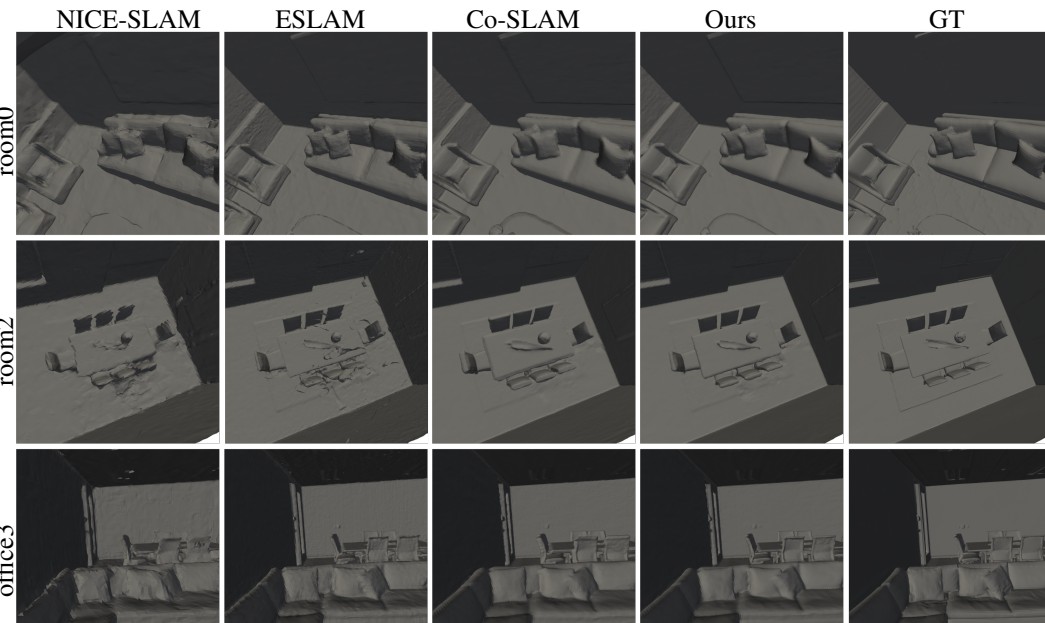

Figure 6: Visualization of Extracted Meshes of Different Methods.

depth data with a certain amount of noise. This demonstrates the robustness of our method when applied to real-world datasets.

### D.3 MORE RESULTS ON MESH EVALUATION

Although the implicit TSDF hash grid no longer contributes to the map after a final Gaussian ellipsoid density query, we still retained this intermediate variable to generate the mesh of the relevant scene and compare its completeness with other methods. Tab. 13 quantitatively compares our method with other methods in terms of Accuracy (in cm), Completion (in cm), and Completion Ratio (in %). Fig. 6 shows the mapping metrics of our method compared with other methods (primarily NeRF-based methods, as they allow for mesh extraction) on the Replica dataset.

### D.4 MORE RESULTS ON GAUSSIAN ELLIPSOID DISTRIBUTION

In Fig. 7, we present the distribution of ellipsoid distances from the object surface in GSS (a) and our system (b). Thanks to the constraints of the TSDF hash grid, the ellipsoids in our system are more concentrated on the object surface.

Table 11: Statistical Significance Analysis on the Scannet dataset

| Methods | 0000 | 0059 | 0106 | 0169 | 0181 | 0207 |
|---|---|---|---|---|---|---|
| Ours | 0.23 | 0.41 | 0.15 | 0.17 | 0.22 | 0.19 |

Table 12: More Results on Recent Open-sourced 3DGS-based Systems

| Methods | \multicolumn{9}{c|}{Replica} | \multicolumn{4}{c}{Tum} |
|---|---|---|---|---|---|---|---|---|---|---|---|---|---|
| | r0 | r1 | r2 | o0 | o1 | o2 | o3 | o4 | Avg. | 1desk | 2xyz | 3office | Avg. |
| Photo-SLAM | 0.54 | 0.39 | 0.30 | 0.52 | 0.44 | 1.28 | 0.78 | 0.58 | 0.60 | 2.60 | **0.34** | **1.00** | 1.31 |
| RTG-SLAM | 0.20 | 0.18 | 0.13 | 0.22 | **0.12** | 0.22 | 0.20 | **0.19** | 0.18 | 1.66 | 0.38 | 1.13 | **1.06** |
| GS-ICP | **0.15** | **0.16** | **0.11** | **0.18** | **0.12** | **0.17** | **0.16** | 0.21 | **0.16** | 2.70 | 1.80 | 2.70 | 2.40 |
| Ours | 0.27 | 0.30 | 0.34 | 0.22 | 0.27 | 0.28 | 0.37 | 0.29 | 0.29 | **1.44** | 1.01 | 1.53 | 1.33 |

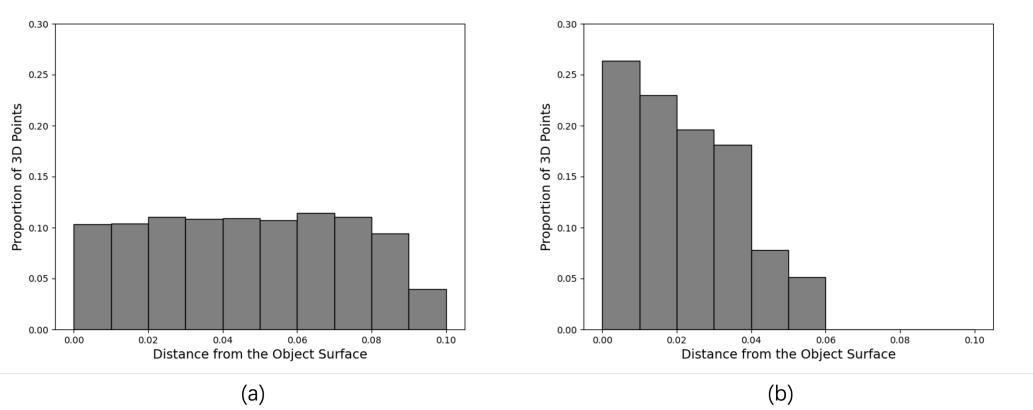

(a)  (b)

Figure 7: Histogram of the distance distribution of Gaussian ellipsoids from the surface.

Table 13: Results on Mesh Evaluation

| Methods | Metric | R0 | R1 | R2 | O0 | O1 | O2 | O3 | O4 | Avg. |
|---|---|---|---|---|---|---|---|---|---|---|
| TSDF-Fusion | Acc.↓ | 4.21 | 3.08 | 2.88 | 2.70 | 2.66 | 4.27 | 4.07 | 3.70 | 3.45 |
| | Comp.↓ | 5.04 | 4.35 | 5.40 | 10.47 | 10.29 | 6.43 | 6.26 | 4.78 | 5.12 |
| | Comp. Ratio (%)↑ | 76.90 | 79.87 | 77.79 | 79.60 | 71.93 | 71.66 | 65.87 | 77.11 | 75.09 |
| iMAP | Acc.↓ | 4.14 | 3.16 | 3.96 | 3.37 | 2.13 | 4.18 | 4.28 | 4.53 | 3.71 |
| | Comp.↓ | 5.99 | 4.57 | 5.23 | 3.91 | 3.97 | 4.89 | 5.65 | 6.81 | 5.12 |
| | Comp. Ratio (%)↑ | 77.84 | 85.39 | 79.02 | 83.01 | 88.05 | 79.17 | 73.42 | 74.29 | 80.02 |
| NICE-SLAM | Acc.↓ | 2.48 | 2.14 | 2.21 | 1.87 | 1.62 | 3.32 | 3.05 | 2.58 | 2.41 |
| | Comp.↓ | 2.68 | 2.23 | 2.85 | 1.92 | 1.86 | 3.23 | 3.27 | 3.72 | 2.72 |
| | Comp. Ratio (%)↑ | 91.66 | 93.42 | 91.32 | 94.80 | 93.94 | 88.12 | 87.53 | 87.08 | 90.98 |
| Co-SLAM | Acc.↓ | 2.03 | 1.60 | 1.95 | 1.47 | 1.27 | 2.74 | 3.01 | 2.41 | 2.06 |
| | Comp.↓ | 2.07 | 1.86 | 1.99 | 1.63 | 1.64 | 2.48 | 2.77 | 2.50 | 2.13 |
| | Comp. Ratio (%)↑ | 95.16 | 95.19 | 93.48 | 96.09 | 94.55 | 91.63 | 90.62 | 90.32 | 93.38 |
| ESLAM | Acc.↓ | 2.52 | 2.51 | 1.76 | 1.61 | 1.98 | 2.87 | 2.53 | 2.14 | 2.24 |
| | Comp.↓ | **1.98** | 1.78 | 1.78 | 1.50 | **1.41** | 2.05 | **2.27** | **2.28** | **1.88** |
| | Comp. Ratio (%)↑ | **96.04** | 95.09 | 95.88 | 97.32 | **96.66** | 94.38 | **94.01** | **93.10** | **95.31** |
| Ours | Acc.↓ | **1.90** | **1.52** | **1.66** | **1.36** | **1.19** | **2.36** | **2.46** | **2.00** | **1.80** |
| | Comp.↓ | 2.15 | **1.71** | **1.57** | **1.41** | 1.51 | **1.97** | 2.33 | 2.48 | 1.89 |
| | Comp. Ratio (%)↑ | 95.43 | **95.82** | **96.42** | **97.77** | 96.16 | **94.72** | 93.45 | 91.65 | 95.12 |

