# OpenReview forum: "Geometrically Constrained Gaussian Splatting SLAM"
_ICLR.cc/2025/Conference — ICLR 2025 Conference Withdrawn Submission_

### Official Review · Reviewer_1Jwo · 2024-10-25

**Soundness:** 3
**Presentation:** 2
**Contribution:** 2
**Rating:** 5
**Confidence:** 3

**Summary:**

This paper combines the implicit and explicit representation by introducing Truncated Signed Distance Function (TSDF) hash grid to constrain the distribution of Gaussian ellipsoids. This methodology enhances both the quality of the reconstruction and the accuracy of the tracking process.

**Strengths:**

This paper conducts extensive experiments and shows that their method achieves state-of-the-art tracking and mapping accuracy with high efficiency.

**Weaknesses:**

1) I appreciate the authors' integration of TSDF with a 3DGS-based SLAM framework. However, this approach feels somewhat incremental, given that TSDF has already been employed in NeRF-based SLAM. It would be valuable to include an analysis comparing the advantages of this method against modified 3DGS techniques [1].

2) A notable limitation of 3DGS-based SLAM methods is the efficient detection of loop closures, which remains unaddressed in this paper. While the paper emphasizes improvements in tracking, it does not provide visual results demonstrating loop closure. Although I do not expect this method to surpass established techniques such as photo-SLAM or other traditional SLAM-based modules, omitting loop closure limits the scope of innovation and reduces the paper's overall impact.

3) Regarding Figures 4 and 6, it would be beneficial to include emphasized or highlighted areas, as shown in Figure 1. The current figures do not demonstrate advantages in rendering quality or geometric accuracy over other 3DGS-based methods.

[1] Lu, Tao, et al. "Scaffold-gs: Structured 3d Gaussians for view-adaptive rendering." CVPR. 2024.

**Questions:**

According to Figure 2 in SAD-GS [1], directly using render depth loss may fail to provide correct surface reconstruction, I wonder if the author can do some ablation study to determine whether the two depth losses in your formula (9) provide positive optimization effects.

[1] Kung, Pou-Chun, et al. "SAD-GS: Shape-aligned Depth-supervised Gaussian Splatting." CVPRW. 2024.

---

### Official Review · Reviewer_owgn · 2024-11-02

**Soundness:** 2
**Presentation:** 2
**Contribution:** 1
**Rating:** 3
**Confidence:** 5

**Summary:**

This paper presents an RGB-D SLAM approach that uses a hybrid representation combining implicit TSDF and explicit 3D Gaussians. The method leverages SDF as a smoothed geometry estimator to initialize Gaussians. It demonstrates competitive results on benchmark datasets.

**Strengths:**

- Hybrid map representation combining implicit and explicit methods.
- Competitive benchmark performance in tracking, mapping, memory consumption, and runtime analysis.

**Weaknesses:**

- Limited technical novelty. The method is a straightforward integration of existing components; MonoGS (3DGS system), Co-SLAM (hashgrid), and ESLAM (TSDF supervision).
- Integrating grid and point representations in a simple way can weaken the unique strengths of each approach. Point-based representations, for example, allow flexible, on-demand resource allocation without the need for predefined scene boundaries or resolutions. However, introducing a neural field can reduce this flexibility.
- Despite these limitations in novelty, the method offers only a marginal insight—namely, that smoothed geometry provides better initialization of Gaussian locations. The method is an engineering case study rather than a technical contribution to ML community.

**Questions:**

Why is memory usage lower than Co-SLAM? This method requires both 3D Gaussian and hash-grid encoding, which would suggest a higher memory requirement.

---

### Official Review · Reviewer_j4sd · 2024-11-04

**Soundness:** 2
**Presentation:** 2
**Contribution:** 2
**Rating:** 3
**Confidence:** 4

**Summary:**

The authors introduce a novel SLAM approach that leverages a hybrid 3D representation by integrating an implicit, multi-resolution TSDF hash grid with explicit Gaussian primitives. This combination enriches the traditional Gaussian-based SLAM pipeline by enhancing mapping performance, which, in turn, improves pose accuracy. Experimental results demonstrate the method's effectiveness, highlighting its potential in advancing SLAM performance.

**Strengths:**

1. The manuscript is clear and easy to understand.
2. The results demonstrate the effectiveness of the hybrid 3D representation with minimal additional time overhead.

**Weaknesses:**

The authors lack a deep understanding of basic concepts such as Gaussian representations, mapping, and localization, which results in superficial and imprecise descriptions throughout the manuscript, particularly in the Introduction. This lack of rigor affects the overall clarity and logical flow of the paper.

1. The authors consistently use the term “Gaussian ellipsoid” throughout the manuscript, which lacks rigor and accuracy. A Gaussian function represents a probabilistic distribution, characterizing the influence or effectiveness of a primitive within its surrounding region through a soft representation anchored at the primitive center. Using the term "ellipsoid" to describe a Gaussian primitive is misleading and stems from a simplification of Gaussian visualizations. I recommend revising this terminology to more precisely reflect the Gaussian function's role in soft spatial representation, rather than suggesting a fixed, ellipsoidal form.
2. In lines 37–39, the authors suggest that Gaussian Splatting (GS) is suboptimal for representing 3D geometric structures, an assertion that is intuitively understandable. However, the manuscript would benefit from a clearer explanation of how this limitation affects pose estimation. Specifically, the impact of GS on pose accuracy seems more indirect compared to its effect on mapping. I recommend that the authors conduct an in-depth analysis, supported by experiments, to examine how the hybrid map representation directly enhances mapping accuracy and, by extension, indirectly improves localization performance. A comparison of mapping and localization outcomes with and without the hybrid representation would offer valuable insights.
3. The authors currently assess mapping performance using depth error alone, which provides only a limited perspective on mapping improvements. Given the hybrid 3D representation proposed in this work, depth error alone does not sufficiently capture the contribution of the TSDF representation to mapping accuracy. A more comprehensive evaluation metric, such as mesh accuracy or 3D reconstruction quality, would more effectively demonstrate the mapping enhancements introduced by the TSDF component. Including these metrics would provide a clearer validation of the hybrid approach’s impact on spatial fidelity and mapping robustness.
4. In lines 46–47, using submaps is one approach to address the large number of Gaussian splats, but not the only one; the statement should be revised for accuracy.
5. The ablation study should include mapping performance and present additional 3D visualizations to better illustrate the effects of the proposed method.
6. The manuscript lacks the visualization for the proposed TSDF representation.

**Questions:**

1. It would be valuable for the authors to discuss how the proposed method compares to existing works that transfer 3D Gaussian representations directly into mesh or other representations better suited to capturing surface characteristics. Many state-of-the-art approaches (such as SuGaR) can transform Gaussian splats into mesh or similarly structured representations, which often enhances geometric detail and surface fidelity. The manuscript would benefit from clarifying the unique advantages or improvements this hybrid representation offers compared to these transformation-based approaches. Specifically, it would be helpful to see a comparison in terms of accuracy, computational efficiency, or other aspects that highlight this method’s distinct contributions.
2. In Figure 1, why is it difficult to ascertain the correctness or improvement in the Gaussian representation within the highlighted red box regions, as the ellipsoid visualizations appear similar between the baseline and the proposed method? Additionally, what accounts for the significant black hole in the GSSLAM representation in the second example?

---

### Comment · Reviewer_1Jwo · 2024-11-26

The authors did not provide a rebuttal to address the raised concerns. As a result, I keep my original score.

---

### Note · Authors · 2024-11-26

**Comment:**

Thank you to all the reviewers for their valuable feedback and constructive comments. After careful consideration, we have decided to withdraw this paper. We greatly appreciate the insightful suggestions provided by reviewers 1Jwo, j4sd, and owgn, and these will be incorporated into the revised version of the manuscript.

**Withdrawal Confirmation:**

I have read and agree with the venue's withdrawal policy on behalf of myself and my co-authors.